# Surface Plasmon Resonance Assay for Label-Free and Selective Detection of HIV-1 p24 Protein

**DOI:** 10.3390/bios11060180

**Published:** 2021-06-03

**Authors:** Lucia Sarcina, Giuseppe Felice Mangiatordi, Fabrizio Torricelli, Paolo Bollella, Zahra Gounani, Ronald Österbacka, Eleonora Macchia, Luisa Torsi

**Affiliations:** 1Dipartimento di Chimica, Universita’ degli Studi di Bari A. Moro, 70125 Bari, Italy; lucia.sarcina@uniba.it (L.S.); paolo.bollella@uniba.it (P.B.); luisa.torsi@uniba.it (L.T.); 2CNR–Institute of Crystallography, 70125 Bari, Italy; giuseppe.mangiatordi@ic.cnr.it; 3Dipartimento di Ingegneria dell’Informazione, Università degli Studi di Brescia, 25123 Brescia, Italy; fabrizio.torricelli@unibs.it; 4Physics, Faculty of Science and Engineering, Åbo Akademi University, 20500 Turku, Finland; zahra.gounani@abo.fi (Z.G.); ronald.osterbacka@abo.fi (R.Ö.); 5CSGI (Centre for Colloid and Surface Science), 70125 Bari, Italy

**Keywords:** HIV-1 p24 protein, surface plasmon resonance, surface modifications, label-free detection

## Abstract

The early detection of the human immunodeficiency virus (HIV) is of paramount importance to achieve efficient therapeutic treatment and limit the disease spreading. In this perspective, the assessment of biosensing assay for the HIV-1 p24 capsid protein plays a pivotal role in the timely and selective detection of HIV infections. In this study, multi-parameter-SPR has been used to develop a reliable and label-free detection method for HIV-1 p24 protein. Remarkably, both physical and chemical immobilization of mouse monoclonal antibodies against HIV-1 p24 on the SPR gold detecting surface have been characterized for the first time. The two immobilization techniques returned a capturing antibody surface coverage as high as (7.5 ± 0.3) × 10^11^ molecule/cm^2^ and (2.4 ± 0.6) × 10^11^ molecule/cm^2^, respectively. However, the covalent binding of the capturing antibodies through a mixed self-assembled monolayer (SAM) of alkanethiols led to a doubling of the p24 binding signal. Moreover, from the modeling of the dose-response curve, an equilibrium dissociation constant K_D_ of 5.30 × 10^−9^ M was computed for the assay performed on the SAM modified surface compared to a much larger K_D_ of 7.46 × 10^−5^ M extracted for the physisorbed antibodies. The chemically modified system was also characterized in terms of sensitivity and selectivity, reaching a limit of detection of (4.1 ± 0.5) nM and an unprecedented selectivity ratio of 0.02.

## 1. Introduction

One of the main features of a biosensing platform is combining a high sensitivity with selectivity in the binding interactions between immobilized biorecognition species and the target analyte [1,2]. Relevantly, the design of a high throughput and reliable transducing interface in biosensors plays a pivotal role in the positive outcome of the assay. Indeed, the immobilization of bioreceptors to a surface always results in the reduction or loss of mobility. Consequently, to prevent any partial or complete loss of bioactivity, arisen from random orientation or structural deformations, bioreceptors should be attached onto surfaces without affecting conformation and functions. Indeed, the biosensor analytical figures of merit might be strongly influenced by the parameter related to the immobilization process itself [3,4].

Many efforts have been made to study suitable immobilization techniques of biorecognition elements on metal surfaces [5,6,7,8]. Some advantages may arise from the stable anchoring of biomolecules by covalent immobilization by forming chemical bonds between complementary functional groups present on the biomolecules and on the solid surface, compared to their direct adsorption on sensor surfaces [3]. For instance, by using antibody fragments or protein G mediated immobilization, a more efficient capture of the bio-recognition element has been observed, thus improving the sensitivity of immunosensing platforms [9,10]. On the other hand, physical immobilization is particularly suited to deposit biorecognition elements on various surfaces. Indeed, it does not require any additional coupling reagents or chemical modification of the biomolecules, therefore being cost-effective and faster than other immobilization techniques. Nevertheless, the resulting biofilms usually lack homogeneity, and the long-term stability of the device needs to be assessed [11]. In the present work, HIV p24 antibodies (anti-p24) by physisorption and chemical deposition through self-assembled monolayers on a 0.42 cm^2^ wide gold detecting interface were characterized with surface plasmon resonance (SPR) for the first time. In particular, the detection efficacies toward human immunodeficiency virus (HIV-1) p24 capsid proteins were compared utilizing the SPR real-time monitoring of the bio-affinity reactions.

The HIV-1 p24 protein is one of the most important biomarkers for the timely and accurate diagnosis of HIV infection due to its presence in the serum or plasma as early as 4–11 days after infection, while only by weeks 3–12 of infection do the HIV host antibodies generally become detectable [12,13]. Therefore, tests that detect the p24 antigen generally allow for the timely detection of HIV infection than the ones based on host antibodies to HIV [14]. Remarkably, blood serum from individuals recently infected with HIV contains from 10 to 30,000 virions per mL, resulting in an estimated concentration of the p24 capsid antigen in the femtoMolar range (fM, 10^−15^ M) [15]. The study of new platforms for the early detection of HIV infection, through an anti-p24 biofunctionalized detecting interface, is of great interest [16], especially from the perspective of developing disposable tests, suitable as fast screening platforms, in the early stage of infection [17,18,19].

To this aim, multi-parameter SPR is herein proposed for the real-time study of biological interaction occurring at the biofunctionalized detecting surface [20,21] as well as a reliable and label-free detection method, achieving limits of detection comparable to the label-needing enzyme-linked immunosorbent assay (ELISA) gold standard [22]. In particular, the binding affinity constants were evaluated for both the immobilization strategies, achieving an equilibrium dissociation constant K_D_ of 5.30 × 10^−9^ M for the assay performed on the SAM modified surface, compared to a K_D_ of 7.46 × 10^−5^ M for that with physisorbed antibodies. This evidence suggests a reduced ligand affinity for the physiosorbed anti-p24 binding sites. Remarkably, the selectivity of the SPR assay in the presence of interferent species has been evaluated. Notably, the human C-reactive protein (CRP) was cross-tested for the first time, demonstrating the selectivity of the immunosensor for p24 detection, achieving an unprecedented selectivity ratio—computed as the ratio between the SPR angle-shifts—as low as 0.02. Moreover, a limit of detection (LOD) of (4.1 ± 0.5) nM was also demonstrated, falling in the same range of the LOD gathered with the label-needing ELISA gold standard and being one order of magnitude lower than the state-of-the-art limit of detection reported for HIV-1 p24 direct SPR assays. Remarkably, this study provides important pieces of information for a reliable and optimized biofunctionalization strategy suitable for further developing a wide-field bioelectronic sensor [19] to accomplish an efficient pre-symptomatic diagnosis of diseases caused by HIV infections.

## 2. Materials and Methods

Mouse monoclonal antibodies to HIV-1 p24 (anti-p24) and the recombinant HIV-1 p24 capsid protein (p24, molecular weight 26 kDa), expressed in *Escherichia coli*, were purchased from Abcam (Cambridge, UK). Human C-reactive protein (CRP, molecular weight 118 kDa) was purchased from Sigma-Aldrich-Darmstadt, Germany. 3-Mercaptopropionic acid (3MPA) (98%), 11-mercaptoundecanoic acid (11MUA), ethanolamine hydrochloride (EA), 1-ethyl-3-(3-dimethylamino-propyl)carbodiimide (EDC), N-hydroxysulfosuccinimide sodium salt (NHSS), and bovine serum albumin (BSA, molecular weight 66 kDa) were purchased from Sigma-Aldrich and used without further purification. A phosphate buffered saline (PBS, phosphate buffer 0.01 M, KCl 0.0027 M, NaCl 0.137 M, Sigma-Aldrich) tablet was dissolved in 200 mL HPLC water and used upon filtration on a Corning 0.22 µm polyethersulfone membrane. 2-(N-morpholino)ethane-sulfonic acid (MES) was purchased from Sigma-Aldrich; a 0.1 M buffer solution was prepared and adjusted at pH 4.8–4.9 with sodium hydroxide solution (NaOH 1 M).

The glass sensor slides (SPR Navi-200) were provided with a 50 nm gold layer on a 2 nm layer of chromium adhesion promoter. They were used after a dip cleaning in a NH_3_ aq./H_2_O_2_ aqueous solution (1:1:5 *v*/*v*) at 80–90 °C for 10 min, then rinsed with water, dried with nitrogen, and treated for 10 min in a UV–ozone cleaner.

A BioNavis-200 multi-parameter surface plasmon resonance (MP-SPR) NaviTM instrument, in the Kretschmann configuration, was used. The SPR modulus was equipped with two laser sources (at 670 and 785 nm wavelengths) and they were both set at 670 nm, scanning an angular range of 50.29–77.93 degrees (SPR liquid range). A single channel cell was used, provided with an internal volume of 100 µL. The injections in the SPR cell were made manually with a 1 mL sterile syringe, with no automatic flow-rate setting.

All data were analyzed with Origin2018 graphing software by OriginLab Corporation.

## 3. Results and Discussion

### 3.1. Gold Layer Bio-Modification

The characterization of both physiosorbed and covalently immobilized anti-p24 capturing antibodies on a gold surface was assessed via surface plasmon resonance (SPR) characterization. To this aim, a multi-parameter SPR (MP-SPR) Navi 200-L apparatus in the Kretschmann configuration was used [23]. In Figure 1, a schematic of the apparatus is shown. Two laser beams, inspecting two different sample areas, pass through the high refractive index material (the prism), and are totally reflected at the low refractive index material (i.e., at the prism–metal layer interface) [23]. The presence of the noble metal thin film (50 nm gold layer) causes partial loss of the reflected light by exciting the metal surface electrons. This produces an evanescent wave that propagates along the interface between the dielectric (sample medium) and the metal layer [24,25]. The so-called surface plasma wave is mostly confined at the metal–dielectric boundary and decreases exponentially into both media, with higher field concentration in the dielectric [25]. Thus, the technique is very sensitive to any variation in the local refractive index on this surface, where biomolecule interactions can be inspected.

The scanning of the SPR resonance angle allows the real-time monitoring of each specimen approaching the metal surface from the dielectric medium. Hence, this setup was used first to characterize the efficacy of the immobilization strategies proposed to deposit the capturing antibodies on the detecting surface, and then to study the interaction of the bio-recognition element with its cognate ligands (vide infra). The biofunctionalized SPR slide holding an area of 0.42 cm^2^ was inspected in two different spots by two laser sources, both set at 670 nm (green and orange arrows in Figure 1) to estimate the layer homogeneity.

Two immobilization strategies were compared for the biofunctionalization of gold electrodes with the anti-p24 antibodies against the HIV-1 p24 capsid protein. The former consists of a physisorption of the bio-recognition elements directly on bare gold; the latter involved the chemical modification of the surface utilizing self-assembled monolayers (SAMs) of alkylthiols.

The biolayer formation on surfaces can be monitored with SPR in real-time. Thus, as shown in Figure 2, the physisorption of anti-p24 was performed in situ by scanning the plasmon peak angle vs. time. Once the baseline was established in PBS, the solution of anti-p24 antibodies at a concentration of 50 µg/mL was injected into the cell. The contact with the surface was kept for two hours, after which the equilibrium was reached between molecules deposited on gold and those in the bulk solution [26]. Then, by rinsing the cell with PBS, the unbounded residues of anti-p24 are removed. The angular shift (∆θ_SPR_) recorded upon anti-p24 physisorption is reported on the sensogram in Figure 2. Moreover, the subsequent deposition of BSA (100 μg/mL in PBS) on the same surface has been performed to prevent non-specific binding.

The surface coverage of physisorbed anti-p24 can be determined by using Feijter’s Equation (1) [27,28]:Γ = d ∙ (n − n_0_) ∙ (*d*n/*d*C)^−1^(1)
where Γ, expressed in ng cm^−2^, is the surface coverage; d is the thickness of the biolayer deposited on the gold surface; (n − n_0_) is the difference between the refractive index of the layer and the one of the bulk medium; and *d*n/*d*C is the so-called refractive index increment of the adsorbed biolayer [28]. Deriving the equation further to consider the instrument response, the difference in refractive index returns Equation (2):
(n − n_0_) = ∆θ_SPR_ ∙ k(2)
where Δθ_SPR_ is the experimental angular shift, and k is the wavelength dependent sensitivity coefficient. For laser beams with λ = 670 nm and thin layers (d < 100 nm), the following approximations hold true: (i) *d*n/*d*C ≈ 0.182 cm^3^ g^−1^, (ii) k∙d ≈ 1.0∙10^−7^ cm∙deg [29].

Therefore, under these assumptions, and by including Equation (2) in Equation (1), the surface coverage Γ can be expressed as a function of the experimental angular shift (Δθ_SPR_, deg) [7,30]:
Γ = Δθ_SPR_·550 (ng/cm^2^),(3)
which can also be expressed in the number of molecules per cm^2^, by considering the molecular weight of the species.

The average value of ∆θ_SPR_, measured from the two curves shown in Figure 2, was used to calculate the surface coverage of the physisorbed anti-p24 antibodies. An experimental value of ∆θ_SPR_ = (0.33 ± 0.04) deg was registered for the physisorbed anti-p24 antibodies. Hence, by using Equation (3), the surface coverage was calculated, obtaining an average value of (181 ± 20) ng/cm^2^, corresponding to (7.5 ± 0.3) × 10^11^ molecules/cm^2^. Additionally, the BSA blocking step on the sensor surface determined a further variation in the SPR angle, with a shift of ∆θ_SPR_ = 0.059 ± 0.002 deg, corresponding to (1.2 ± 0.3) × 10^11^ molecules/cm^2^ BSA molecules. The resulting values for the surface coverage of anti-p24 and BSA layers are reported as the average among the surface coverages evaluated on two different replicates and four different sampled areas. The error bars were estimated as the relative standard deviation.

On the other hand, the chemical bonding of anti-p24 antibodies on the detecting interface foresees the chemical modification of the gold surface with a self-assembled monolayer (SAM) of mixed alkanethiols, prior to the bio-layer formation [30,31]. The mixed SAM with different chain lengths is preferable for anchoring large biomolecules like antibodies, since it provides improved accessibility for protein binding due to reduced steric hindrance [32]. To this aim, gold-coated glass slides were immersed, immediately after cleaning, in the thiol solution. A mixture of 11MUA: 3MPA (1:10 molar ratio) in ethanol was used at a final concentration of 10 mM. The sample, immersed in the thiol solution, was left overnight in a nitrogen atmosphere at room temperature. Afterward, the slide was rinsed in ethanol and mounted in the SPR sample holder.

To achieve the bio-conjugation of antibodies on the SAM, the established EDC/NHSS coupling method was used [33,34]. A scheme of the procedure is depicted in Figure 3 and extensively discussed elsewhere [30]. Briefly, the carboxylic terminal groups of the chemical SAM are converted into intermediate reactive species (NHSS, N-hydroxysulfosuccinimide esters) that react with the amine groups of the antibody, anti-p24, at a concentration of 50 µg/mL, achieving its covalent coupling. Then, the ethanolamine saturated solution (EA, at concentration 1 M) is injected to deactivate the unreacted esters in an inactive hydroxyethyl amide. Finally, to cover possible voids on the SAM and to prevent non-specific binding, a BSA solution 100 μg/mL in PBS was used [35].

The real-time monitoring of the anti-p24 chemical bonding on the mixed-SAM is reported in the sensogram of Figure 4a, showing the variation in the SPR signal for each biofunctionalization step. It is worth mentioning that any possible change in the refractive index due to the buffer composition could lower the SPR signal. To this end, to make sure to control any possible effect due to the different refractive index of the solvent involved in the biofunctionalization, a stable baseline has been recorded before each step. Specifically, for the covalent binding of the bio-recognition elements, the baseline level was established in PBS, as indicated from the bottom-up arrows in the sensogram. The succeeding injections of ethanolamine and BSA were also performed in the same buffer. In Table 1, all the details on the reagents injected and the time of exposure are reported as well as the angular shift, ∆θ_SPR_, measured after the binding of anti-p24, the deactivation with EA, and the adsorption of BSA.

As reported in Table 2, the contact of the activated SAM with the antibodies was kept for two hours, the time required to observe the saturation of the bio-recognition elements, which reached equilibrium on the SAM modified surface. The biolayer homogeneity was also assessed for this immobilization strategy. Thus, the SPR angular shift is reported as the average signal of four replicate experiments while the error bars were evaluated as the relative standard deviation. The sensogram portion corresponding to the anti-p24 binding is highlighted in Figure 4b. Here, the variation in the SPR signal with a value of 0.13 ± 0.02 deg can be appreciated.

Once the bio-conjugation is completed, the surface is exposed to the ethanolamine solution to deactivate the unreacted sites on the SAM. A decrease of 15% in the angular shift was measured after EA had been registered, with the ∆θ_SPR_ equal to 0.11 ± 0.02 deg. This is ascribable to the removal of unreacted anti-p24 antibodies after EA injection.

This experimental value can be used to determine the surface coverage of anti-p24 achieved on the SAM. Hence, by using Equation (3), a coverage of 61 ± 16 ng/cm^2^ was calculated, or equivalently a value of (2.4 ± 0.6) × 10^11^ molecule/cm^2^, being 68% lower than the one achieved with the anti-p24 physisorption.

The following conclusions can be drawn from the comparison of the surface coverage accomplished with the two bio-modification methods. The physisorption of antibodies directly on a gold surface determines the formation of a thick layer of bio-recognition elements, as proven by the surface coverage evaluation. On the other hand, the grafting of antibodies on a chemical SAM gave more control on the number of sites on which the antibodies could be attached, in agreement with previous studies on the modification of gold surfaces using amine coupling on mixed SAMs reported elsewhere [31,32,34,36].

### 3.2. SPR Binding of p24 Proteins to the Anti-p24 Modified Gold Slides

The binding efficacy of the bio-recognition elements was evaluated against HIV-1 p24 capsid proteins (p24) for both physisorption and covalent binding immobilization strategies [37,38].

The analysis was focused on the kinetics of the SPR response registered upon p24 exposure of the physisorbed antibodies or the covalently bound ones. SPR binding of p24 proteins was registered according to the following protocol. The assay was carried out by recording the baseline in PBS and injecting p24 solutions in PBS at different concentrations ranging from 5 × 10^−10^ M to 1 × 10^−6^ M (the detailed sensograms are reported in Appendix A, respectively). Each solution was let to interact with the functionalized interface for 40 min. Upon equilibrium, the protein excess was removed, by rinsing the cell with the PBS buffer solution.

The angular shift, ∆θ_SPR_, was calculated for each concentration as the difference between the equilibrium value after rinsing with PBS and the initial baseline. Thus, in Figure 5, the dose-curves for the assayed protein are reported as ∆θ_SPR_ vs. [p-24] nominal concentrations (semi-log scale). Here, the response measured for the binding of p24 on physisorbed anti-p24 is shown as blue circles, while the binding with anti-p24 conjugated with the chemical-SAM is reported as red squares. Remarkably, an enhancement in the SPR signal of 65% was registered for the p24 detection occurring on the SAM modified surface, as shown in Figure 5. Indeed, although physical adsorption endows the detecting interface with a higher number of deposited capturing antibodies, it likely reduces the availability of antibody active sites, thus resulting in the lowering of the SPR response upon exposure to the antigen solution [39].

The kinetic analysis of the experimental data was performed according to Hill’s binding model for both assays [40]. The solid lines in Figure 5 are the result of the fitting against the Hill equation:(4)$$Y=V_{max}\cdot \frac{X^{n}}{k^{n}+X^{n}}$$
which describes the dependence of the assay response at equilibrium, *Y* = Δθ_SPR_, from the analyte concentration, *X* = [p24].

The equation returns the Hill parameter, *n*, and the apparent dissociation constant, k, (i.e., the analyte concentration corresponding at half of the maximum response (*V_max_*) or, equivalently, at half occupied binding sites) [40,41]. In Equation (4), the term k^n^ represents the equilibrium dissociation constant (*K_D_*) of the binding pairs, which estimates the analyte/antibody binding affinity [41]. Moreover, the Hill parameter, *n*, reflects the degree of cooperativeness of the target molecules interacting with the available binding sites: *n* = 1 holds for a non-cooperative binding while, *n* > 1 and *n* < 1 apply for positive and negative cooperativity, respectively [42,43]. The fitting parameters found for the two assays are reported in Table 3, along with the calculated K_D_.

By applying Hill’s model, a value of K_D_ = 5.3 × 10^−9^ M was estimated when the p24 proteins bind to the chemically grafted anti-p24, which is in excellent agreement with previously reported results [44]. Indeed, the affinity binding constant of anti-p24 antibodies vs. p24 showed a K_A_ = 1/K_D_ value falling in the 10^8^–10^9^ M^−1^ range [45]. On the other hand, the K_D_ calculated for the p24 dose-curve on physisorbed anti-p24 (blue trace of Figure 5) gave a value of 7.46 × 10^−5^ M. This result evidences a lower binding affinity between the physisorbed bio-recognition element and the analyte. By applying Hill’s model, it is worth mentioning that if incompletely bound species accumulate at the detecting interface, the equation could fail to provide physicochemically correct equilibrium concentrations and/or interaction parameters [40,42]. This holds true, especially when the Hill coefficient diverges from *n* = 1, and there is no extreme positive cooperative effect among the binding pairs, leading to an overestimated K_D_ value [40]. Thus, because of the lower Hill coefficient, *n* = 0.64 ± 0.01 (Table 3), obtained for the physisorbed system, some effects of the incomplete binding pairs at the sensor surface cannot be ruled out [40].

The biosensing assay, comprising the covalently bound antibodies, was further characterized in terms of selectivity and sensitivity. Indeed, the anti-p24 modified surface through chemical SAM was tested against the exposure to a non-binding protein, the human C-reactive protein (CRP). In Figure 6, the SPR responses of the antigen p24 (in red) and the non-binding CRP (in black) are shown. Both assays were performed in the same experimental conditions (vide infra). The CPR solutions in PBS at increasing concentration in the range from 5 × 10^−10^ M to 1 × 10^−6^ M were kept in contact with the modified surface for 40 min. Then, upon equilibrium, the SPR cell was rinsed with PBS and the relevant angle-shift (∆θ_CRP_) was measured.

As observed in Figure 6a, the selectivity of the biosensing platform was successfully demonstrated. Indeed, the negative control experiment showed a maximum angle-shift below 0.01 deg, being only 3% of the signal registered for the p24 assay. Accordingly, the selectivity of the assay was estimated as the ratio between the angle-shift measured for CRP and p24 binding, respectively [46]. The resulting value was as low as ∆θ_CRP_/∆θ_p24_ = 0.01/0.46 = 0.02, which demonstrated extremely high selectivity performances [47].

The limit of detection of the assay was also evaluated. To this aim, the linear portion of the calibration curve of p24 in linear scale was considered (Figure 6b). In the same figure, the black squares are the data of the negative control experiment involving CRP. Thus, the LOD was calculated as the average signal of the negative control experiment (s_CRP_) plus three times its standard deviation (σ_CRP_). This signal was as high as y = s_CRP_ + 3σ_CRP_ = 8 × 10^−3^ deg. Hence, the comparison of this level with the interpolating linear regression of Figure 6b resulted in a LOD of (4.1 ± 0.5) nM. The LOD found in this study was one order of magnitude lower than direct SPR detection methods, which reached limits of detection most at 40 nM, depending on several factors such as the experimental configuration, the sample’s optical property, and binding affinity of target molecules [20,22,47,48].

Additionally, to compare the performances of the two biofunctionalization protocols, the gold surface with physisorbed anti-p24 was further characterized, with the same method used for the SAM modified surface. In Figure 7, the SPR responses of the antigen p24 (in blue) and the non-binding CRP (in black) are shown. The physisorbed antibodies were exposed to the non-binding protein, CRP, in the range of concentration between 5 × 10^−9^ M and 6 × 10^−7^ M. The contact of the solutions with the surface was kept for 40 min, after which PBS was used to rinse the protein excess. The relevant angle-shift (∆θ_CRP_) was measured and compared with the response of the p24 protein. As observed in Figure 7a, the selectivity of the biosensing platform could be demonstrated. The maximum angle shift measured for the cross-reaction was below 0.02 deg, with 4% of the signal coming from the analyte. Nevertheless, the ratio between the angle-shift measured for CRP and p24 binding, respectively, resulted in a value of ∆θ_CRP_/∆θ_p24_ = 0.01/0.13 = 0.08, which implies a slightly lower selectivity compared to the chemically modified surface.

Then, to evaluate the LOD of the assay performed on the physisorbed anti-p24, the linear portion of the calibration curve of p24 in linear scale was considered (Figure 7b, blue circles). In the same figure, the value measured for the CRP assay is depicted as black squares and their average value (s_CRP_) as a black dotted line. Thus, the LOD was calculated as the average signal of the negative control experiment (s_CRP_) plus three times its standard deviation (σ_CRP_). This signal was as high as y = s_CRP_ + 3σ_CRP_ = 2.3 × 10^−2^ deg. Hence, the comparison of this level with the interpolating linear regression of Figure 7b resulted in a LOD of (27 ± 1) nM. Relevantly, the LOD found for the SAM modified surface was one order of magnitude lower than that calculated for the physisorbed anti-p24.

The main analytical figures of merit of the two biofunctionalization methods are summarized in Table 4. The percentage relative standard deviation (RSD%) was estimated over three independent experiments on nominally identical experimental conditions. The LOD of the assays was calculated from the linear plot of the SPR response of p24 over the SAM-binding and the physisorbed antibody, respectively, as shown in Figure 6b and Figure 7b (vide infra). The sensitivity was determined from the same plot as the slope of the linear fit. The selectivity is expressed as the ratio between the angular shift measured for the control experiment with CRP and the response of the analyte, p24, at the highest concentration assayed.

## 4. Conclusions

In conclusion, the characterization of two biofunctionalization strategies for gold surfaces was performed through a multi-parameter SPR assay. The physisorption of antibodies against HIV-1 p24 (anti-p24) directly on the bare gold detecting surface led to the immobilization of (7.5 ± 0.3) × 10^11^ molecule/cm^2^. The covalent binding of anti-p24 on a mixed SAM of alkanethiols brings a decreased surface coverage of (2.4 ± 0.6) × 10^11^ molecule/cm^2^, thus being 68% lower than the one registered with physisorbed capturing antibodies.

However, the chemical immobilization endows the detecting interface with a reduced steric hindrance between the closest neighbor biorecognition elements, providing enhanced capturing efficacy toward the target analyte. Indeed, a doubled response was recorded for the latter assay. In addition, compared to the physisorbed antibodies, the covalently bound anti-p24 also resulted in a lower dissociation constant. In fact, K_D_ values of 7.46 × 10^−5^ M and 5.30 × 10^−9^ M were measured, respectively, highlighting a better analyte/antibody binding affinity for the assay on anti-p24 modified SAM.

The biosensing assay of both covalently bound and physisorbed anti-p24 were also characterized in terms of selectivity and sensitivity. The modified surfaces were tested against the exposure to a non-binding protein, the human C-reactive protein (CRP), for the first time and the response was compared to the p24 signal in the same range of concentrations. This allowed for the estimation of a selectivity ratio as low as 0.02, and a limit of detection of (4.1 ± 0.5) nM for the covalently bound antibodies, being one order of magnitude lower than the state-of-the-art limit of detection of 40 nM reported for the direct SPR assays and comparable to the label needing ELISA gold standard. This study thus represents a proof of principle of the early detection of HIV infection. Meanwhile, a selectivity ratio of 0.08 and a limit of detection of (27 ± 1) nM were found for the physisorbed anti-p24 assay. Moreover, this SPR characterization could pave the way toward developing reliable bio-electronic platforms in which the gold sensing electrode can be modified following the biofunctionalization strategy assessed in the present study.

## Figures and Tables

**Figure 1 biosensors-11-00180-f001:**
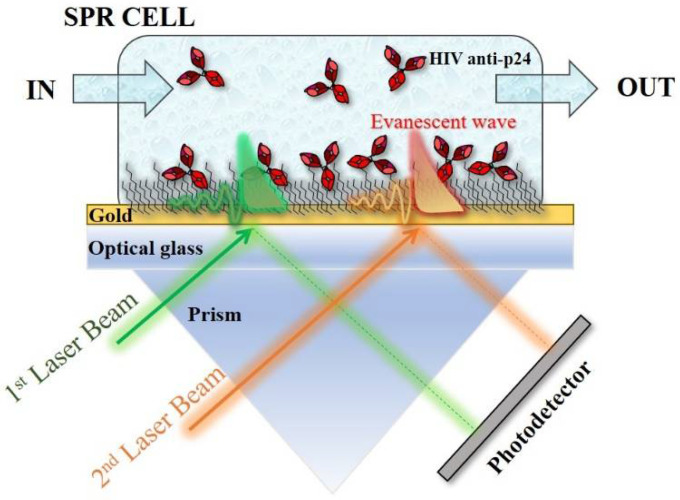
SPR apparatus in the Kretschmann configuration. During the biofunctionalization of gold, the green and orange laser beams (λ = 670 nm) sampled the surface in two points and serves to monitor the layer homogeneity.

**Figure 2 biosensors-11-00180-f002:**
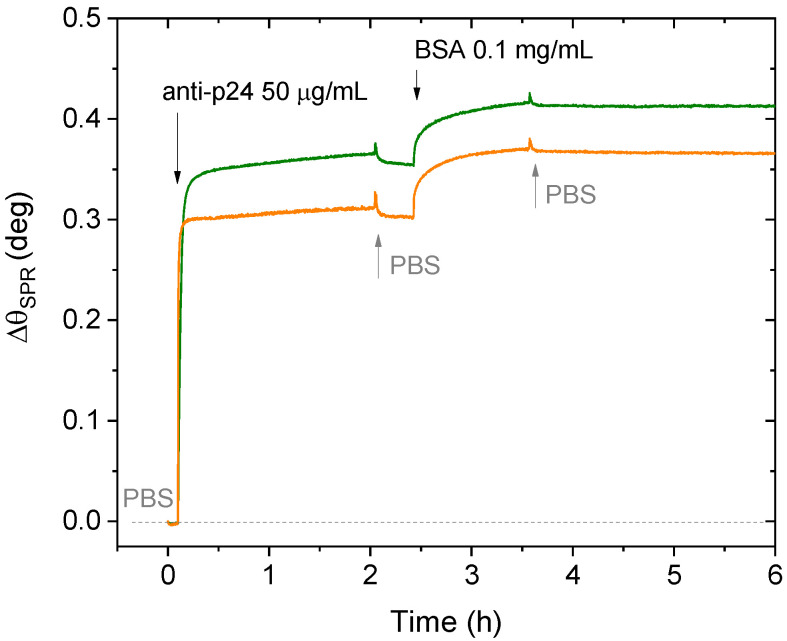
SPR sensogram of the anti-p24 physisorption on the gold surface and the subsequent BSA blocking. Green and orange curves refer to the surface inspection performed by the two laser beams in two different points of the sample points.

**Figure 3 biosensors-11-00180-f003:**
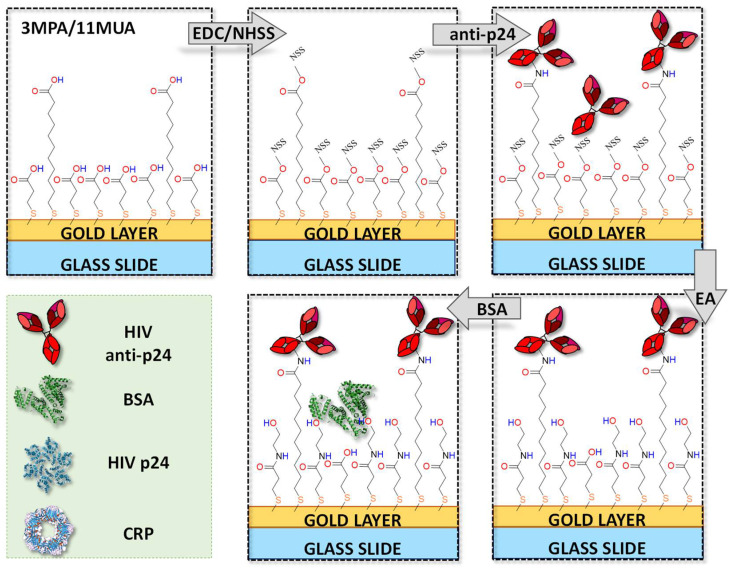
Schematic representation of the anti-p24 covalent bonding on the mixed SAM (3MPA/11MUA) on gold (not in scale). The inset on the left side reports a legend of all the biorecognition elements and analytes involved in the sensing and negative control experiments.

**Figure 4 biosensors-11-00180-f004:**
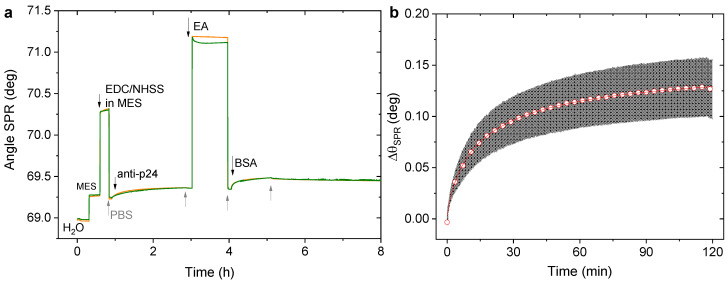
(**a**) SPR sensogram of the anti-p24 covalent immobilization through mixed-SAM on the gold SPR slide. Two area were sampled on the surface in each experiment, shown as green and orange curves. (**b**) Detail of the anchoring of anti-p24, reported as an average signal over four replicate experiments (red circle) along with their standard deviation (grey shadow).

**Figure 5 biosensors-11-00180-f005:**
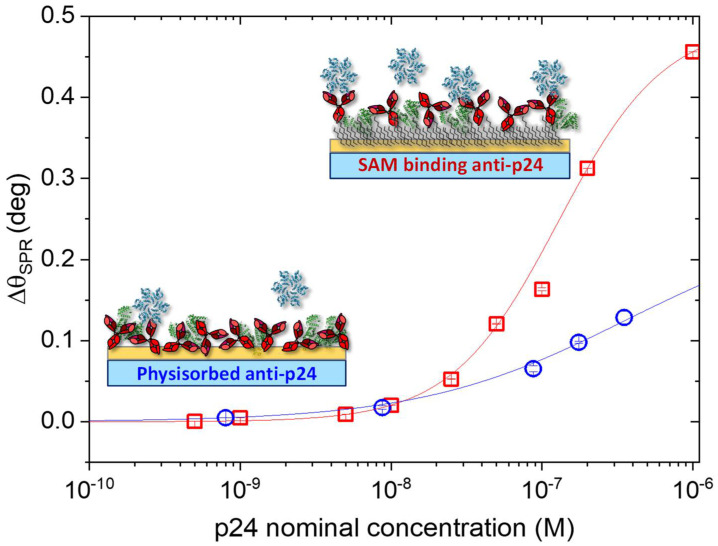
SPR response for the p24 binding on anti-p24 covalently bound on the chemical SAM (red squares) and for the p24 binding on physisorbed anti-p24 (blue circles). The Hill fitting model is shown as red and blue solid lines, respectively.

**Figure 6 biosensors-11-00180-f006:**
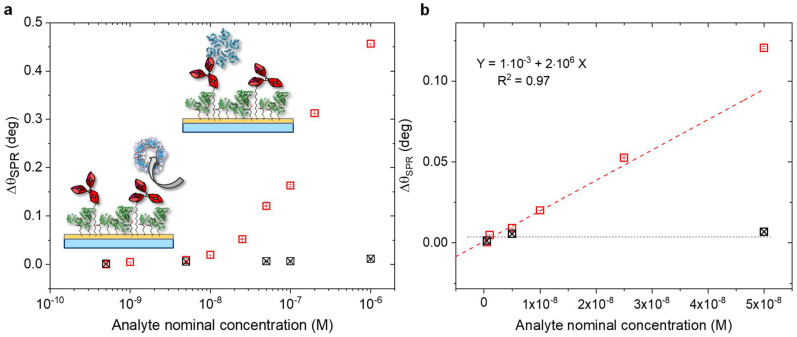
(**a**) SPR angular shift vs nominal concentration of HIV-1 p24 (red squares) and CRP (black squares) in the cross-reactivity test, performed on the SAM modified surface (semi-log scale). (**b**) Linear plot of SPR response performed on modified anti-p24 SAM, upon the p24 (red squares) and CRP (black squares) binding vs. analyte nominal concentration. The regression of the linear portion of p24 response is shown as the red dotted line; the average signal of the negative control is depicted as the black dotted line. The average value of three replicate analysis and their standard deviation are reported.

**Figure 7 biosensors-11-00180-f007:**
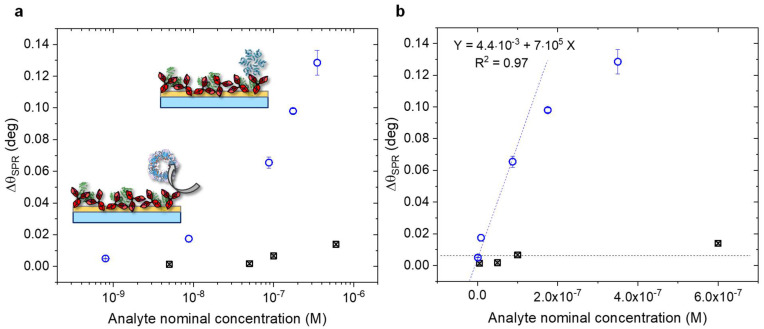
(**a**) SPR angular shift vs nominal concentration of HIV-1 p24 (blue circles) and CRP (black squares) in the cross-reactivity test, performed on physisorbed antibodies (semi-log scale). (**b**) Linear plot of SPR response performed on the modified surface, upon the p24 (blue circles) and CRP (black squares) binding vs, analyte nominal concentration. The regression of the linear portion of p24 response is shown as a blue dotted line; the average signal of the negative control is depicted as a black dotted line. The average value of three replicate analysis and their standard deviation are reported.

**Table 1 biosensors-11-00180-t001:** Differences in performance between multi-parameter SPR and ELISA gold standard for the detection of HIV-1 p24 proteins.

p24 Detection Method	Detection-Type	Limit of Detection	Assay Steps	Label-Needing	Assay Time
**ELISA** [22]	quantitative	40 nM	5	yes	at least 5 h
***MP-SPR***	quantitative	4 nM	2	no	<1 h

**Table 2 biosensors-11-00180-t002:** Experimental condition used for the modification of the activated mixed-SAM in the SPR apparatus. The reagent composition, time of exposure, and SPR response is reported for each biofunctionalization step.

	Reagent	Time	∆θ_SPR_ (deg)
Antibodies conjugation	anti-p24 (50 µg/mL) in PBS	2 h	0.13 ± 0.02
Bond-saturation	EA (1 M) in PBS	45 min	^1^ 0.11 ± 0.02
Blocking	BSA (100 μg/mL) in PBS	1 h	0.08 ± 0.01

^1^ The angular shift is relevant to the anti-p24 bond after EA deactivation.

**Table 3 biosensors-11-00180-t003:** The parameters obtained from the Hill fitting are reported along with their standard error for both immobilization methods.

Hill Fit	Vmax	k	*n*	R^2^	k^n^ = K_D_ (M)
SAM-binding	0.492 ± 0.004	(1.27 ± 0.03)∙10^−7^	1.2 ± 0.03	0.999	5.30∙10^−9^
Phys-anti-p24	0.25 ± 0.01	(3.6 ± 0.2)∙10^−7^	0.64 ± 0.01	0.998	7.46∙10^−5^

**Table 4 biosensors-11-00180-t004:** Summary of the analytical figures of merit of the two biomodification methods with anti-p24 for the detection of the p24 protein.

	RSD (%), n = 3	LOD (nM)	Sensitivity (deg·M^−1^)	Selectivity (Δθ_CRP_/ Δθ_p24_)
**SAM-binding**	1.0	4.1 ± 0.5	(1.9 ± 0.2)·10^6^	0.02
**Phys-anti-p24**	6.2	27 ± 1	(7 ± 2)·10^5^	0.08

## Data Availability

The data presented in this study are openly available in https://www.fairdata.fi/en/ (accessed on 2 June 2021). Repository Research data storage service IDA (ida.fairdata.fi).

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
