# Peer review of "Surface Plasmon Resonance Assay for Label-Free and Selective Detection of HIV-1 p24 Protein"

_biosensors, 2021, doi:10.3390/bios11060180_

Round 1

Reviewer 1 Report

The manuscript “Surface Plasmon Resonance assay for label-free and selective detection of HIV-1 p24 Protein” investigated the label-free detection of HIV-1 p24 capsid protein by two-channel SPR instrument, through both the physisorbed and covalent binding of p24 antibodies on the gold surface of the SPR chip. The results show that the covalent binding method is able to detect p24 protein at a limit of 4.1 ± 0.5 nM at selectivity of 0.02. The manuscript is well written with clear descriptions, thus I would recommend its publication on Biosensors with following revisions.

  1. An experiment should be added to check the non-specific bonding experiment with CRP for the physisorbed p24 antibodies modified surface. This is because the paper is for comparison of physisorbed and covalent binding for p24 detection, the selectivity comparison for these two methods is significant.
  2. The LODs both for physisorbed and covalent binding methods should be given.
  3. The authors claim the covalent binding method has high sensitivity, the authors should provide a comparison table for other existing methods such as ELISA for HIV-1 p24 measurements.
  4. The authors should also introduce for clinic use, what is the sensitivity required. Data for experiments with p24 spiked blood (or serum, or BSA) should be included in order to claim that this bioassay is useful for clinical use.

Author Response

Referee: 1

The manuscript “Surface Plasmon Resonance assay for label-free and selective detection of HIV-1 p24 Protein” investigated the label-free detection of HIV-1 p24 capsid protein by two-channel SPR instrument, through both the physisorbed and covalent binding of p24 antibodies on the gold surface of the SPR chip. The results show that the covalent binding method is able to detect p24 protein at a limit of 4.1 ± 0.5 nM at selectivity of 0.02. The manuscript is well written with clear descriptions, thus I would recommend its publication on Biosensors with following revisions.

Issue #1: An experiment should be added to check the non-specific bonding experiment with CRP for the physisorbed p24 antibodies modified surface. This is because the paper is for comparison of physisorbed and covalent binding for p24 detection, the selectivity comparison for these two methods is significant.

Reply: We fully agree with the Reviewer and we have edited the Discussion Section of the manuscript accordingly. In particular, Figure 7 to show the negative control experiment with a physisorbed anti-p24 modified SPR slide toward CRP. The selectivity performance of the physisorbed slide has been compared to the one achieved for the covalent binding immobilization strategy.  

Issue #2: The LODs both for physisorbed and covalent binding methods should be given.

Reply: We acknowledge the Reviewer for rinsing this important point. The LOD has been computed also for the physisorbed SPR slide and the regression of the linear portion of p24 response is shown in Figure 7b.

Issue #3: The authors claim the covalent binding method has high sensitivity, the authors should provide a comparison table for other existing methods such as ELISA for HIV-1 p24 measurements.

Reply: We acknowledge the Reviewer for rinsing this important point. For the sake of clarity, we have introduced Table 1 at the end of the Introduction Section to highlight the differences in performance between Multi-parameter SPR and state-of-the-art platforms for detection of HIV1 p24 protein.

Issue #4: The authors should also introduce for clinic use, what is the sensitivity required. Data for experiments with p24 spiked blood (or serum, or BSA) should be included in order to claim that this bioassay is useful for clinical use.

Reply: We thank the Reviewer for rising such an important point. Remarkably, blood serum from individuals recently infected with HIV contains from 10 to 30.000 virions per mL, resulting in estimated concentration of the p24 capsid antigen in the femtoMolar range (fM, 10-15 M).[D.M. Rissin et al., Nature Biotechnol. 2010, 20(6), 595-599.] Therefore, improving the sensitivity of the biosensing platform might result in a timely diagnosis of HIV infection, with public health benefits. We fully agree with the Reviewer about the importance of repeating the experiments in real biofluids. However, further studies will be carried on by the authors to prove the detection of p24 proteins in real biofluids such as blood serum, breast milk, semen and vaginal secretions. While a real fluid containing the HIV-1 p24 protein has not being yet assayed by the authors, nonetheless the PBS solution used reproduces a physiologically relevant fluid with a pH of 7.4 and ionic strength of 162 mM, mimicking the environment of blood serum.

Reviewer 2 Report

The paper by Sarcina et al. describes a biosensor for the detection of p 24 protein, which is a component of  the HIV virus capsid; the presence of p24 in blood plasma or serum indicates HIV infection. The biosensor consists of anti-p24 antibody immobilised onto a gold surface via mercaptoundecanoic acid linker and the EDS/NHS protocol. Under model conditions, the developed biosensor shows linearity of the analytical response over the 5 -50 nM range . Parallel experiments with  the anti-p24 antibody immobilised by adsorption onto a chip surface were performed with negative results. Surface coverage was determined, and the fit  of the results to the Hill model was investigated. Considering  that the developed biosensor is intended  finally to be an analytical tool for  HIV detection, surprisingly few analytical characteristics were determined. Therefore, major revision is recommended.

Specific comment:

  1. What is an the expected level of p24 concentration in blood serum or plasma?
  2. Precision and recovery of the biosensor should be determined.
  3. An example of p24 determination in blood should be given, at minimum with plasma or serum spiked with p24
  4. Fig S3 should be transfered from supplementary material to the main body of the manuscript.
  5. NH4OH should be replaced by NH3 aq.

Author Response

Referee: 2

The paper by Sarcina et al. describes a biosensor for the detection of p24 protein, which is a component of the HIV virus capsid; the presence of p24 in blood plasma or serum indicates HIV infection. The biosensor consists of anti-p24 antibody immobilised onto a gold surface via mercaptoundecanoic acid linker and the EDS/NHS protocol. Under model conditions, the developed biosensor shows linearity of the analytical response over the 5 -50 nM range . Parallel experiments with the anti-p24 antibody immobilised by adsorption onto a chip surface were performed with negative results. Surface coverage was determined, and the fit  of the results to the Hill model was investigated. Considering that the developed biosensor is intended  finally to be an analytical tool for  HIV detection, surprisingly few analytical characteristics were determined. Therefore, major revision is recommended.

Issue #1: What is the expected level of p24 concentration in blood serum or plasma?

Reply: We acknowledge the Reviewer for rinsing this important point. It has been recently demonstrated that blood serum from individuals recently infected with HIV contains from 10 to 30.000 virions per mL, resulting in estimated concentration of the p24 capsid antigen in the femtoMolar range (fM, 10-15 M) [D.M. Rissin et al., Nature Biotechnol. 2010, 20(6), 595-599.]. Taking the Reviewer’s comment into account, this aspect has been added in the main text at lines 65-68, page 2.

Issue #2: Precision and recovery of the biosensor should be determined.

Reply: We acknowledge the Reviewer for rinsing this important point. The authors addressed the Reviewer comment introducing Table 4 in the discussion section, in which the main analytical figures of merit of the assay have been reported for both biofunctionalization strategies. The precision has been estimated as the agreement between three independent test results obtained under the specified experimental conditions for the biomodification methods. The percentage relative standard deviation (RSD%) has been reported to quantify the precision of the biosensor. [Andreasson, Ulf et al. Frontiers in neurology 2015, vol. 6 179. 19 Aug. doi:10.3389/fneur.2015.00179] Following the methods reported in this work, the biosensor could be successfully modified with the antibodies for the HIV-1 p24 protein. Eventually the determination of the recovery of the sensor could be performed for spiked biological matrix taking advantage of the calibration curve authors performed in buffered solutions of p24 protein. However, this study aimed at the assessment of the biomodification method, and the assay of non-hazardous p24 samples, in prospect of further assays of real samples.

Issue #3: An example of p24 determination in blood should be given, at minimum with plasma or serum spiked with p24.

Reply: We acknowledge the Reviewer for his/her valuable comment and we agree on the importance of real biofluids assay. However, further studies will be carried on by the authors to prove the detection of p24 proteins in real biofluids such as blood serum, breast milk, semen and vaginal secretions. While a real fluid containing the HIV-1 p24 protein has not being yet assayed by the authors, nonetheless the PBS solution used reproduces a physiologically relevant fluid with a pH of 7.4 and ionic strength of 162 mM, mimicking the environment of blood serum.

Issue #4: Fig S3 should be transfered from supplementary material to the main body of the manuscript.

Reply: We agree with the Reviewer and we have edited Figure 6 accordingly.

Issue #5: NH4OH should be replaced by NH3 aq.

Reply: We apologize for the inconsistences that has been corrected.

Round 2

Reviewer 1 Report

The authors have addressed all of my comments, and I recommend the publication of this paper in its present form. 

Just one typo, in line 364, "platform can be demonstrate" should be "platform can be demonstrated".

Author Response

We acknowledge the Reviewer for having spotted this typo. We have corrected the typo.

We thanks once again the Reviewer for his/her suggestions and support.

Reviewer 2 Report

The authors supplemented manuscript with the sentence:

Remarkably, blood serum from individuals recently infected with HIV contains from 10  to 30.000 virions per mL, resulting in estimated concentration of the p24 capsid antigen in  the femtoMolar range (fM, 10-15 M) [15].

On the other hand, LOD of the developed the Multi-parameter SPR is 2 nM  (Table 1) i.e. significantly below fM range. Such conclusion should be added!!!!. Thus minor revision is recommended.

Author Response

We acknowledge the Reviewer for his/her extremely important comments and suggestions. We are glad the Reviewer appreciated our work.

We have know added a sentence in the conclusion, according to Reviewer's suggestion.